# Intraurban Geographic and Socioeconomic Inequalities of Mortality in Four Cities in Colombia

**DOI:** 10.3390/ijerph20020992

**Published:** 2023-01-05

**Authors:** Laura A. Rodriguez-Villamizar, Diana Marín, Juan Gabriel Piñeros-Jiménez, Oscar Alberto Rojas-Sánchez, Jesus Serrano-Lomelin, Victor Herrera

**Affiliations:** 1Department of Public Health, School of Medicine, Universidad Industrial de Santander, Bucaramanga 681012, Colombia; 2School of Medicine, Universidad Pontificia Bolivariana, Medellin 050031, Colombia; 3School of Public Health, Universidad de Antioquia, Medellin 050010, Colombia; 4Division of Public Health Research, Project Bank Team, National Institute of Health-INS Colombia, Bogotá 111321, Colombia; 5Department of Public Health, Queen’s University, Kingston, ON K7L 3N6, Canada

**Keywords:** mortality, poverty, health disparities, spatial analysis, cardiovascular diseases, respiratory tract diseases, neoplasms, Colombia

## Abstract

Mortality inequalities have been described across Latin American countries, but less is known about inequalities within cities, where most populations live. We aimed to identify geographic and socioeconomic inequalities in mortality within the urban areas of four main cities in Colombia. We analyzed mortality due to non-violent causes of diseases in adults between 2015 and 2019 using census sectors as unit of analysis in Barranquilla, Bogotá, Cali, and Medellín. We calculated smoothed Bayesian mortality rates as main health outcomes and used concentration indexes (CInd) for assessing inequalities using the multidimensional poverty index (MPI) as the socioeconomic measure. Moran eigenvector spatial filters were calculated to capture the spatial patterns of mortality and then used in multivariable models of the association between mortality rates and quintiles of MPI. Social inequalities were evident but not consistent across cities. The most disadvantaged groups showed the highest mortality rates in Cali. Geographic inequalities in mortality rates, regardless of the adults and poverty distribution, were identified in each city, suggesting that other social, environmental, or individual conditions are impacting the spatial distribution of mortality rates within the four cities.

## 1. Introduction

Mortality profiles reflect changes associated with the epidemiologic transition and differences in the access to healthcare services across the socioeconomic gradient in a territory, revealing inequalities between regions and countries [1]. Since 1950, an increase in life expectancy at birth of around 23 years on average has been observed globally for both males and females [2], and there are significant variations within and between countries that reveal territorial and socioeconomic inequalities [3,4,5]. The access to healthcare services is also closely related to mortality profiles, as many deaths can be attributed to conditions that can be mitigated in the presence of effective healthcare [6]. Being the most robust health outcome in the majority of countries, the analysis of mortality allows governments to identify medium- and long-term infrastructure and knowledge needs, as well as plann the allocation of resources [1].

The World Health Organization’s Sustainable Development Goals call for reducing unfair differences in health outcomes, especially in the most inequitable populations, such as those living in Latin American countries. In the last 25 years, most cities in this region have experienced changes in the trends of overall and cause-specific mortality as they have transitioned from communicable to non-communicable diseases, which is a pattern partially explained by improvements in populations’ income, education, and access to public health programs [6]. Although mortality rates showed a declining trend in the region after 2000, with a significant gain in life expectancy at birth, there are few changes in the differences across countries, gender and age groups, as an expression of remaining health inequalities [2,7]. The later has also been observed within urban areas, specifically in the capital cities of Argentina, Brazil, Chile, Costa Rica, Mexico, and Panama, where spatial disparities in life expectancy correlate with intra-urban differences in educational level and socioeconomic status [8].

In Colombia, six of the top ten causes of mortality are related to non-communicable diseases [2], which are led by cardiovascular diseases (32%), neoplasms (26%), and chronic respiratory diseases (20%); however, in contrast to cities, violence and traffic accidents remain the main causes of death in rural areas (44%) [9]. In this country, inequities in mortality from non-communicable diseases, mainly cardiovascular diseases and neoplasms, have consistently been described across geopolitical units (departments and municipalities) as a function of education [10,11,12,13] and other socioeconomic indexes, such as the unsatisfied basic needs (UBN), the human development index (HDI), the housing deficit index (HDI), and the multidimensional poverty index (MPI) [14,15,16]. According to these observations, higher proportions of women, young, and uneducated individuals constitute the main drivers of inequity in mortality. Similar results have recently been reported for mortality from external causes and COVID-19 [17,18].

Environmental and socioeconomic attributes are important determinants of health at the intra-urban level; however, the richness and high heterogeneity of most Latin American cities’ landscapes translates into complex, difficult to unveil spatial patterns [19]. A recent analysis of mortality across 363 cities in the region showed that, although much of its variability is explained by differences between countries, city-specific factors also seem to play an important role [6]; however, little is still known about mortality inequalities at a smaller spatial scale [20]. The identification of intra-urban spatial and socioeconomic patterns of health inequalities is not only relevant considering that about three out of four people in Colombia reside in cities [9], but also constitutes the first step to understand and tackle the problem through area-specific public health policies. In this study, we aimed to identify spatial and socioeconomic inequities in the overall and cause-specific mortality at small geographical scale within the urban areas of the four largest cities in Colombia.

## 2. Materials and Methods

### 2.1. Study Design and Population

We conducted an ecologic retrospective study of mortality due to non-violent causes of diseases in adults with a focus on socioeconomic and geographical inequality in four main capital cities in Colombia between 2015 and 2019 (See Appendix A). Colombia is the country located at the extreme north of South America with an estimated population of 50 million in 2019 [21]. People living in urban areas account for 77.1% of the total population, and the largest cities are Bogotá (the capital district with 8,000,000 in population), Medellín (2.8 million in population), Cali (2.5 million in population) and Barranquilla (750,000 in population). Bogotá is located at the center of the country, Medellín is located at the northwest, Cali is located at the southwest, and Barranquilla is located at the north in the Caribbean coast. The study population included all residents in the urban area of the four cities who died between 1 January 2015 and 31 December 2019. We included non-violent deaths of adults over 18 years old.

### 2.2. Definition of Geographical Areas

We used census sectors (CS) as geographical unit of analysis. The CS are intermediate census geographical areas defined by the National Administrative Department of Statistics (DANE, for its acronym in Spanish) in Colombia for census purposes. The CS were measured in aggregate blocks, which are the smallest geographical unit for which the census population information is available. The CS were aggregated into census sections which corresponded to broader areas that could include different neighborhoods within the urban area. Thus, the census sector is the smallest geographical census unit for which population and census data are available with a low risk of identification of cases (deaths). The population size varies across CS within and between cities, so they are not homogeneous units in terms of population. We used the cartographic information and maps from the 2018 National Census from the DANE Geoportal public website [22]. The spatial data were created in ArcGIS 10.8.1^®^ using the projection of Colombia in modes custom azimuth equidistant and datum WGS 1984.

### 2.3. Data Sources

#### 2.3.1. Socioeconomic and Population Data

We used the Colombian multidimensional poverty index (MPI) as a measure for assessing socioeconomic inequalities. The MPI is a composite index that has been designed, calculated, and supplied by DANE since 2011. The MPI represents the measurement of poverty from the multi-deprivation point of view. The MPI calculation is based on five dimensions (education level; conditions of children and youth; employment; health; and public utilities access and housing conditions). Each dimension has a value of 0.2 points (20%) and all together compile 15 indicators with a specific weight based on the number of indicators per dimension. The health dimension includes health coverage and access but does not include mortality as part of its indicators. The dimension with the largest number of indicators is access to public utilities and housing conditions (five indicators) [23]. The MPI was built using the Alkire–Foster method, using the household as the unit of analysis. Thus, a household is considered “in a situation of poverty” if its MPI ≥ 33.3%. In the interpretation of the MPI, although the unit of analysis is the household, the deprivations are experienced simultaneously by all its members. A proxy of the national MPI was constructed at the municipal and CS level using census data [24]. We used for this study the Colombian 2018 MPI, as it is located within the span of the study period and represents the more recent MPI measurement using national census data. The index at the municipal and census sector level represents the percentage of the population living in multidimensional poverty, and therefore, the higher the index, the higher the socioeconomic deprivation.

Total population over 18 years, population by sex and age groups, and area of residence (urban/rural) were retrieved at the CS level from the estimations of population based on the DANE census [25]. The population counts and cartographic information for the year 2018 were chosen because this is the most recent census year in Colombia, and all population counts are available at the CS level.

#### 2.3.2. Mortality Data

We obtained mortality data from the National Vital Statistics System, particularly from the Mortality Registry provided by the health authorities of the cities of Barranquilla, Bogotá, Cali, and Medellín. The mortality registry is centralized at the national level by DANE, codified in terms of causes of death, and then made available for all municipalities, including all deaths occurred in the country by place of residency. Thus, the mortality registry for each municipality includes all deaths of its residents that might have occurred outside of their municipality of residence. The mortality registry used the International Classification of Diseases, 10th Revision, ICD-10 diagnostic codes. We included deaths in adults over 18 years old who died from non-violent causes and excluded causes of death with codes ICD-10 S00-T98 and V01-Y98 related to external causes of mortality. Specific causes of death were grouped. for inequality analysis purposes, as deaths due to circulatory diseases (I00-I99), respiratory diseases (J00-J99), and cancer/blood diseases (C00-D48). These three groups of causes of death were chosen, as they correspond to the three main causes of mortality in Colombia [12]. The counts of all deaths and specific causes of death were obtained at the census sector level for the four cities based on the availability of a valid address of residence in the death registry.

### 2.4. Data Analysis

#### 2.4.1. Descriptive Mortality Analysis

Using counts of deaths and population for year 2018 as the average population of the study period (2014–2019), we calculated cumulative crude mortality rates at the CS level. Having small geographical areas, the crude rates might have been unstable due to small counts in some areas, so we calculated smoothed standardized mortality ratios by applying an empirical Bayesian approach [26]. First, the standardized mortality ratio (SMR) was calculated for each CS using the following formulae: SMR = total number of deaths/expected total number of deaths, where the expected number of total deaths (for all causes or specific causes of death) = total population at CS × overall city mortality rate. Second, the SMRs were smoothed (smoothed SMR) using empirical Bayes estimators [27] to account for unstable rates due to low numbers of deaths in some CS. The smoothed SMRs were built using a Poisson random intercept regression model. Using the smoothed SMRs, we calculated Bayesian mortality rates (BMRs) at the CS level. We assessed the spatial autocorrelation of the mortality outcome variables using the Moran’s index. We described mortality counts and BMRs at the CS level using summary measures and maps. We used Stata ^®^ version 13 for calculating smoothed SMR and rates, ESF tool software [28] for calculating Moran’s index, and ArcGIS ^®^ to generate maps.

#### 2.4.2. Spatial Filters

We used the Moran eigenvector spatial filters (MESF) approach [29,30] to (i) identify geographic inequalities of mortality within urban areas of the cities and (ii) to estimate the association between the BMRs and the MPI in a regression model while controlling for covariates and removing potential spatial autocorrelation present in the residuals. The MESF is a relatively novel statistical method that works with the spatial autocorrelation of data to identify spatial patterns in the distribution of a variable (e.g., study outcome) across a geographical area. The method is based on the Moran coefficient of spatial autocorrelation and exhibits statistical properties of unbiasedness, efficiency, robustness and consistency for data with normal, binomial, and Poisson probability distributions [29]. Briefly, the method uses the eigenvector decomposition of a N × N geographical connectivity matrix to extract orthogonal (uncorrelated) numerical components called eigenvectors. Each eigenvector represents an independent pattern (map) that captures the latent spatial autocorrelation of a georeferenced outcome variable at different geographical scales (hyperlocal, intermediate, or more broad scales within the cities). The eigenvectors related to the outcome variable can be linearly combined into a “spatial filter,” which can be used as explanatory variable in a regression model. Thus, spatial filters can be used to observe the geographic patterns of an outcome variable while removing the spatial autocorrelation of the model’s residuals, which results in regression models with better fit and with no violation of assumptions of independence among residuals [29].

We used the BMRs for all deaths and specific causes of death as outcome variables. For the process of obtaining the eigenvectors for each outcome variable, we used a connectivity “queen” matrix for the CS, in which, for each SC, we included CS neighborhoods that shared boundaries on a single node (point) or a segment of border limits. We used only positive spatial autocorrelations and obtained statistically significant (*p* < 0.05) eigenvectors related to BMRs in each city. They were used to build a spatial filter to maximize the explanation of variability for the mortality outcome variable. A spatial filter score was estimated for each CS in all four cities and used in subsequent regression models as the explanatory variable. We used the MESF software ESF tool for extracting the eigenvector solution and creating the spatial filter [28].

#### 2.4.3. Socioeconomic and Geographical Inequality Analysis

We used BMRs for all deaths and specific causes of death at the CS level as the primary outcomes for the inequality analysis. Socioeconomic inequalities were defined as differences in mortality among CS based on the MPI. The geographical inequalities were defined as spatial differences on the mortality rate among CS defined by spatial filters (MESF) for each mortality outcome and city.

##### Concentration Indexes

We estimated the concentration index (CInd) along with 95% confidence intervals (95% CI) and plotted the corresponding concentration curves for the overall and selected specific causes of mortality, with the MPI as a socioeconomic stratification variable and the CS as a unit of analysis, separately for each municipality [31,32]. Then, following Massey et al. [33], we estimated the concentration index at the extremes (CIE) as a measure of spatial social polarization based on block-level MPI within each CS and also separately for each municipality. First, the CIE was calculated for each CS as the ratio of the difference between the number of blocks below the first (most privileged) and above the fifth (most deprived) quintiles of the distribution of the corresponding city’s MPI, and we determined the total number of blocks within the CS. Then, for each municipality, we estimated the CIE as a weighted mean of its values across all CS, with weights equal to the inverse of the ratio between the number of blocks in each CS and the total number of blocks, along with bootstrapped 95% CIs.

#### 2.4.4. Multivariable Inequality Analysis

We used four models to assess the effect of socioeconomic and geographical inequality. In the first model, we assessed the crude association between BMRs and the MPI quintiles with no adjustment for spatial filters. In a second multivariable model, we assessed the first model introducing adjustment for sex (% males) and age (% people 60 years or older). In a third multivariable model, we assessed the second model by introducing the adjustment by the spatial filter. A fourth model was built with the same specification of model 3, but we replaced the quintiles of MPI with the quintiles of ICE as the socioeconomic variable. For models 2–4, we used age and sex adjustments as dichotomous variables and used the third quartile of their distribution as the cut-off value. In all models, we used Poisson regression models using expected counts based on BMRs as the dependent variable and log-transformed population as the off-set. We reported incidence rate ratios (IRR) with a 95% CI for explanatory variables and determined the spatial autocorrelations of residuals using the Moran’s index. We used the Stata 13^®^ and ESF tool software for these analyses. Then, we created quintiles of the spatial filter to define CS with low (Quintile 1) and high (Quintile 5) BMRs for each mortality outcome and city and described the BMRs distribution by the quintiles of spatial filters to visualize geographical inequalities, if present.

We conducted a sensitivity analysis assessing the effect of individual socioeconomic variables, instead of the composite MPI, on the overall non-violent adult mortality in the cities. For this purpose, we used the following socioeconomic variables calculated at the CS level: the percent of adults with primary level or no education as a measure of education, the percent of adults unemployed looking for work as a measure of occupation, and the household strata as a measure of housing conditions. The household strata is a national categorization of household physical conditions developed by DANE that ranges from 1 to 6, with 1 being the most deprived category [34]. All three socioeconomic variables were obtained from data publicly available from the DANE 2018 Census.

## 3. Results

### 3.1. Descriptive Analysis of Mortality and MPI

There were 236,545 non-violent deaths in adults during 2015–2019 in the four cities. The crude median cumulative mortality rates varied across the cities between 2562 in Bogotá and 3603 in Cali. In all cities, the deaths caused by diseases of the circulatory system were the most frequent, followed by deaths related to cancer/blood diseases and respiratory diseases, respectively. Table 1 shows the number of deaths and crude mortality rates for all deaths and specific causes of death at the city and at the census sector level. There was a wide dispersion of the number of deaths at the census sector level within the cities and, therefore, BMRs were calculated. Both crude and smoothed BMRs had similar distribution and median values with differences only existing in the extremes of the distribution. The highest median BMRs were observed in Cali for all adult deaths (median BMR = 3593.99 per 100,000), and the three specific causes of death. The lowest median BMRs at the census sector level was observed in Bogotá (For all deaths BMR = 2560.35 per 100,000 and 2562.53 per 100,000). Figure 1 shows the geographic distribution of the BMRs for all non-violent adult deaths in the four cities. Census sectors (CS) with higher BMRs tended to be clustered towards the center and west in Barranquilla, at the expanded center of the city in Cali and Medellín, and scattered in clusters across the north–south divide in Bogotá. A low, but statistically significant spatial autocorrelation index for the BMRs for all deaths was observed in all cities (Moran’s index: Barranquilla = 0.182 (*p* < 0.001), Bogotá = 0.11 (*p* < 0.01), Cali = 0.224 (*p* < 0.001), and Medellín = 0.215 (*p* < 0.01). Maps of the distribution of BMRs for circulatory, respiratory and cancer/blood causes of death for the cities are presented in Appendix A.

The mean MPI at the census sector level was 8.82 (percentile 25–75 = 3.98–18.26) for Barranquilla, 7.63 (p25–p75 = 4.50–12.56) for Bogotá, 7.73 (p25–p75 = 4.05–13.89) for Cali, and 8.61 (p25–p75 = 4.10–15.59) for Medellín. Figure 2 shows the geographical distribution of MPI quintiles within cities. For Barranquilla, the most socio-economic deprived areas were located at the southeast and west of the city and specific sectors at the north end of the city. In Bogotá, deprived areas were located mainly at the south and east of the city. In Cali, these deprived areas were mainly located at the west and east sides of the city. In Medellín, the sectors with higher MPI were located at the northwest and east of the city. A statistically significant spatial autocorrelation index for MPI was observed in all cities (Moran’s index: Barranquilla = 0.648 (*p* < 0.001), Bogotá = 0.484 (*p* < 0.001), Cali = 0.589 (*p* < 0.001), and Medellín = 0.658 (*p* < 0.001).

### 3.2. Socioeconomic Inequality Measured by CInd and CIE

For all cities and causes of death, the CInd and CIE showed small and negative values, which indicated the presence of discrete disparities with a tendency of the concentration of deaths in the extremes of the CS with higher MPI (Figure 3). Table 2 shows that, for Bogotá and Medellín, the CInd were statistically significant for overall and the three specific causes of non-violent deaths in adults. For Barranquilla, the CInd was statistically significant for overall and cancer mortality, and for Cali, the CInd was statistically significant only for cancer mortality (See Appendix A for circulatory, respiratory, and cancer/blood deaths in the Appendix A).

### 3.3. Regression Models of Socioeconomic Inequality

Table 3 shows the multivariable Poisson regression models on the association between BMRs for all non-violent adult deaths and quintiles of the MPI without and with spatial filters for the four cities during the study period. The crude model (Model 1) and age–sex adjusted model (Model 2) are presented without the inclusion of spatial filters, and a third model and fourth model are presented for MPI (Model 3) and ICE (Model 4) quintiles, respectively, which are adjusted by age, sex, and the spatial filter. The crude models showed some statistically significant differences among MPI quintiles for the cities with lower incidence rate ratios (IRR) for quintiles Q4–Q5 (higher MPI and deprivation) compared to Q1 (lower MPI), particularly in Bogotá and Medellín. In age–sex adjusted models, we found statistically significant lower BMRs in the least privileged quintile for Bogotá and higher BMRs in least privileged quintiles in Cali. An important ecologic effect of the proportion of older populations was observed for all cities. All crude and age–sex adjusted models showed a statistically significant spatial autocorrelation of residuals (Moran I *p*-values = 0.001). When including the spatial filter (see Figure 4) in Model 3, the effects of age–sex variables remained similar to Model 2, and there were positive statistically significant effects of the spatial filter in all cities, suggesting independent effects of geographical patterns from socioeconomic indexes and age–sex structure. Moreover, the Models 3 and 4 with the spatial filter had increased values of pseudo R^2^ compared to the Model 2 for all cities, and their spatial autocorrelation in residuals was removed (Moran I *p*-values > 0.1). These models showed pseudo R^2^ values that ranged between 0.27 in Bogotá and 0.42 in Barranquilla, with only those variables explaining an important proportion of the variability in the BMRs. Models 3 and 4 using MPI and CIE showed similar pseudo-R^2^ values, but ICE seems to better capture the lower IRRs at the least privileged quintiles in Bogotá, the largest city. The Q2–Q3 in Medellín and the Q4 and Q5 in Cali exhibited higher IRRs compared to the most privileged Q1 of the MPI, suggesting higher mortality rates in less privileged quintiles after controlling for age–sex structure and geographical patterns.

Results for Models 3 and 4 for circulatory, respiratory, and cancer/blood deaths are presented in Appendix A. In Barranquilla, there were no statistically significant effects of the MPI or ICE quintiles, but there were significant positive effects of age and spatial filter for the three specific cause of deaths. In Bogotá, in addition to the positive association with age and sex variables, there was a pattern in which the IRR decreased as ICE quintiles increased for circulatory and cancer mortality and reached statistical significance for Q5, suggesting an inequality in mortality for these specific causes. In Cali, in addition to the positive association with age and spatial filter, there was an inverse association with the proportion of the male sex, suggesting higher BMRs in CS with lower proportions of men. Also, for circulatory deaths, the least privileged MPI and ICE quintiles showed higher statistically significant IRRs compared to the most privileged quintiles. In Medellín, in addition to the positive association with age–sex structure and spatial filter, there were statistically significant higher IRRs for Q2–Q3 compared to Q1 for deaths caused by circulatory diseases and lower IRR for Q5 for deaths caused by respiratory and cancer/blood diseases. For all models and cities, there were statistically significant effects of the spatial filter with increased values of pseudo R^2^ and the removal of spatial autocorrelation of residuals when compared to the models without spatial filter.

The spatial filters for each BMR and city were built based on a combination of selected eigenvectors among all possible eigenvectors that described different patterns of positive spatial autocorrelation. The details of the number of selected eigenvectors and the properties of the spatial filter for each city and type of death are presented in Appendix A. Figure 5 shows the distribution of the BMRs for all non-violent adult deaths by quintiles of the spatial filter in the four cities. In all cities, there was an incremental gradient according to the quintiles of the spatial filter that showed higher BMRs in the upper quintiles. Appendix A show the geographic gradient and spatial filter distribution for circulatory, respiratory, and cancer/blood deaths in the Appendix A.

The results of the sensitivity analysis assessing the effect of individual socioeconomic variables, instead of the composite MPI, are presented in Appendix A. For all cities, the spatial filter was statistically significant, which showed a geographical effect on mortality that was independent of socioeconomic variables. The effect of the proportion of older populations on mortality was also consistent across cities. A higher proportion of adults with low education levels (no education or primary level only) was associated with higher BMRs, particularly in Bogotá and Cali. The household strata showed a statistically significant effect in Bogotá, with increased BMRs in the most deprived household strata (strata 1–3) compared to the most privileged (strata 4–6).

## 4. Discussion

Our findings provide evidence of the presence of socioeconomic inequalities with an important contribution of geographical inequalities on non-violent mortality in adults at the intraurban level in four main cities of Colombia. Using a spatial filtering approach, our results suggest the presence of geographical patterns that are associated with mortality and independent of small area socioeconomic conditions. The cities exhibit different characteristics of socioeconomic inequalities, but there is evidence of geographical inequalities in all of them. Up to our knowledge, this is the first study that assesses the effect of socioeconomic and spatial inequities together at intraurban levels in Colombia and South America.

Our results found evidence of socioeconomic inequalities in less privileged populations for overall adults´ non-violent mortality, particularly for Cali when using the MPI as the socioeconomic stratification variable (see Model 3 in Table 3). For this city, the gradients were observed particularly for circulatory causes of death. In contrast, for Bogotá, there was evidence of higher BMRs for overall, circulatory, and cancer/blood diseases in sectors within more privileged quintiles (Q2–Q3) of the MPI (See Appendix A). The MPI is a composite measure of poverty proposed by Akire and Santos [35] that understands the complexity and multidimensional composition of the phenomenon using indicators that go beyond monetary variables. The Colombian MPI was developed by the national government, using a standard international methodology, for the strategic establishment and monitoring of public policies against poverty [23]. The Colombian MPI has been used previously for assessing associations between socioeconomic conditions and the risk of dying from COVID-19 in Colombia at municipal and department level, and has shown positive and significant associations at different points of time during the pandemic [36,37]. The MPI has also been used to assess the association with the prevalence of youth depressive symptoms [38]. The study included comparisons of the composite MPI with income poverty and individual and household measures; their results found associations of depressive symptoms with MPI but not with income poverty as well as associations with education, labor, and access to health deprivation at the individual level but not with household deprivations in Colombia. These findings enforce the need to avoid crossing the level when assessing associations with socioeconomic measures (i.e., combining individual and household or area-level). Our study used BMRs and MPI, both at the CS level, and found a very low contribution of the MPI alone for explaining the variability of overall and specific causes of adult mortalities across sectors.

Our results also found small and negative concentration indexes (CInd) and CI at extremes (CIE) for overall adult mortalities using the MPI as the socioeconomic stratification variable. The CInd is a very well-known tool for measuring health inequalities [32] and the ICE is a relatively novel extension of the CInd for assessing inequalities that has been useful in intraurban settings [39,40]. A similar finding of low and negative CInd was reported by Yanez et al. [41] when assessing the socioeconomic gradients in mortality for cerebrovascular disease at department levels using the unsatisfied basic needs index as a socioeconomic measure (CIns = −0.07) in Colombia between 2014–2016. Despite the unsatisfied basic needs index focusing on poor housing conditions rather than a multidimensional approach, their findings also pointed to a very low gradient in this specific cause of circulatory deaths. We used the CIE as a socioeconomic indicator instead of the MPI in Model 4. The results showed that both MPI and CIE exhibited similar results, but CIE showed more clearly the gradient effect of ICE quintiles in Bogotá and suggested lower BMRs in the higher quintiles compared to the most privileged. Our results suggest that CIE is a useful health inequality metric that can capture intraurban socioeconomic inequalities. Similar findings have been described in North American cities, and the CIE is proposed as an alternative inequality metric for public health monitoring that captures better extremes of deprivation and privilege in socioeconomic variables [39].

We built and compared regression models for BMRs and MPI with sequential adjustment for age and sex structures, and then for the spatial pattern introducing the spatial filter. We found a very low variability in the BMRs which was explained by the MPI and an important explanation of the age and sex structure of the CS, which is expected for natural (non-violent) mortality. In contrast to the MPI, the geographical patterns of mortality at different spatial scales, captured by the spatial filter, provided a higher contribution to the explanation of the variability in the BMRs, particularly in Barranquilla and Cali, where the spatial filter contributed more than 15% of the explained variability compared to the model with the age and sex adjustment. This finding was present for overall mortality and the analysis of specific causes of mortality. The important contribution of the spatial filter in the models shows that the city patterns at the census sector scale are not fully explained by the socioeconomic spatial gradient and that, probably, other geographical conditions at the local level need to be identified as determinants of mortality. Further research needs to address potential conditions related to geographical patterns at individual and neighborhood-sector levels at the cities, and they might involve environmental, behavioral, or physical conditions related to health.

Similar findings related to the high contribution of geographical patterns on morbidity were found by Serrano et al. [42] when analyzing respiratory health service utilization during childhood in two Canadian cities. They used the quintiles of Moran´s eigenvector spatial filters (MESF) to assess the geographical patterns and observed that geographical inequalities in respiratory health service utilization were not completely explained by the spatial distribution of the socioeconomic status measured by material and social deprivation indexes. The MESF approach has been also used to assess spatial inequalities of COVID-19 mortality in England [43]. In this study, the authors aimed to investigate the contributions of socioeconomic and environmental factors to spatial variations in the COVID-19 mortality rate across England and found that, while hospital accessibility was negatively related to COVID-19 mortality rate, the percent of Asians, percent of Blacks, and unemployment rate were positively related to the COVID-19 mortality rate. Similarly, MESF was used in South Korea to assess the spatial pattern of tuberculosis and the socio-environmental factors associated with its incidence rate at administrative unit levels; they found that using the MESF efficiently controlled the residuals´ autocorrelation of the models and showed that the population composition ratio, population growth rate, health insurance payment, and public health variables were significant throughout the study period [44].

Our sensitivity analysis showed that, in models using CS-level measurements of education, occupation, and household conditions separately (instead of the MPI), an inequality gradient in overall non-violent adult mortality was observed for all cities in terms of education, which showed that BMRs increased at the CS level as the percent of adults with primary level or no education increased, particularly for Bogotá and Cali. Our findings are similar to the report of the effect of education on mortality and life expectancy at birth in other cities in Latin America [8]. Similarly, In Bogotá, the effect of household conditions showed increased IRRs for the most deprived strata compared to the most privileged. Our results for the sensitivity analyses also showed consistently that, for all cities, the effect of spatial patterns, measured by the spatial filters, were independent of the socioeconomic variables and produced model´s residuals with no spatial autocorrelation.

The study of social inequalities on overall and specific causes of mortality, particularly cardiovascular and cancer, have been addressed previously at the geographical level in Colombia [13,14,16]. These studies have used ecological approaches with different geographical units that usually correspond to departments or municipalities. The mortality risk, particularly for cardiovascular disease, the most common cause of mortality, has been reported to be higher in small municipalities with high MPI [14]. At the regional level, a recent study conducted in 286 Latin American cities identified predictors of infant mortality in urban settings and found that higher population growth, better living conditions, better service provision, and mass transit availability were associated with lower infant mortality rate, while greater population size was associated with a higher rate [45]. Similar to our findings with adult mortalities, there was no association for population-level, educational attainment, and the infant mortality rate. A similar ecological study in 363 Latin American cities assessed the variation in city-level amenable mortality (the mortality that can be mitigated in the presence of timely and effective health care); their findings showed that population growth and higher city-level socio-economic status were associated with lower amenable mortality [6]. These studies provide a clear geographical large-scale picture of the socioeconomic gradients; however less is known about the profile of the socioeconomic inequalities in intraurban settings.

A recent study conducted in Medellín by Patino et al. [46], assessed the effect of built environment and mortality risk from cardiovascular disease and diabetes during 2016–2017 using the neighborhood as the geographical unit of analysis. They included design and density variables to capture the built environment and included the household strata as the socioeconomic covariate in the models. They found that the average greenness and the fraction of the neighborhood allocated to the highest household strata showed a negative and significant association with all disease’s groups. As per our knowledge, this is the only previous study assessing socioeconomic gradients at an intraurban scale on mortality in a Colombian city, and the non-statistically significant results related to household strata significance observed in our sensitivity analyses for Medellín might be explained by the different way that the household strata variable was handled in the analysis in both studies. Moreover, our study incorporated a spatial filter to capture the spatial pattern and autocorrelation present in this type of geographical ecological analysis, thus allowing for the separate independent effects of socioeconomic variables, as well as the effect of the MPI as a composite measure of poverty.

### Strengths and Limitations

There are strengths of this study that are worthy to mention. First, we used mortality data of high quality and coverage reported by the Vital Statistics System in Colombia [47]. Second, we used the MESF approach for identifying spatial configurations at different geographical scales of the BMRs within the urban areas of the cities. By using this approach, we were able to: (i) capture the geographical configuration of the BMRs within the cities; (ii) identify the effect of MPI as a comprehensive poverty indicator independent of the geographical configuration; and (iii) remove the spatial autocorrelation in model´s residuals making the models free of specification model bias. We included all potential eigenvectors of the MESF solution for the smoothed SMR and did not perform any selection of eigenvectors to capture all possible geographical configurations of mortality at different geographical scales within the cities. Third, we conducted a sensitivity analysis using specific socioeconomic variables at the CS level instead of the MPI to assess the consistency of the results in terms of geographical and socioeconomic inequalities, which allowed us to capture the effect of specific dimensions of the index in the cities, such as education and housing conditions.

It is important to recognize the ecological nature of these analyses and their limitations. Our analyses aimed to assess both geographical and socioeconomic inequality using an ecological approach at a small-area level (census sector). Still, our data is ecological in nature and, therefore, there are many individual-level variables that might be related to mortality that are not available at the ecological scale, such as nutrition, physical activity, and chronic disease control, among others. Those variables may explain the remaining BMR variability that is not explained in our models, including ecological-level variables. However, our study showed that between 27% and 42% of the variability in the BMRs was explained by age and sex structure and social and geographical patterns, which is an important contribution to address mortality gradients. The reduced number of CS as a unit of analysis may have had an effect on the lack of statistical significance of some results, particularly in Barranquilla and Medellín. Finally, the deaths were assigned to the census sector of residency at the time of death, and, therefore, geographical life-course mobility was not accounted in our analysis.

## 5. Conclusions

Our study showed that mortality rates in Barranquilla, Bogotá, Cali, and Medellín were largely and consistently explained by the age population’s structure at the census level. Social inequalities were evident but not consistent across cities. The most socioeconomic disadvantaged groups showed the highest mortality rates only in Cali. By contrast, in Bogota and Medellin, sectors with middle socioeconomic conditions were associated with the highest mortality rates. Geographic inequalities in mortality rates, regardless of the poverty distribution, were identified in each city, suggesting that other social, environmental, or individual conditions were impacting the spatial distribution of mortality rates within the four cities. Our findings may inform current policies and future research aimed at addressing socioeconomic and geographical inequalities at the intraurban level as a strategy to better understand and intervene more efficiently in mortality and its determinants.

## Figures and Tables

**Figure 1 ijerph-20-00992-f001:**
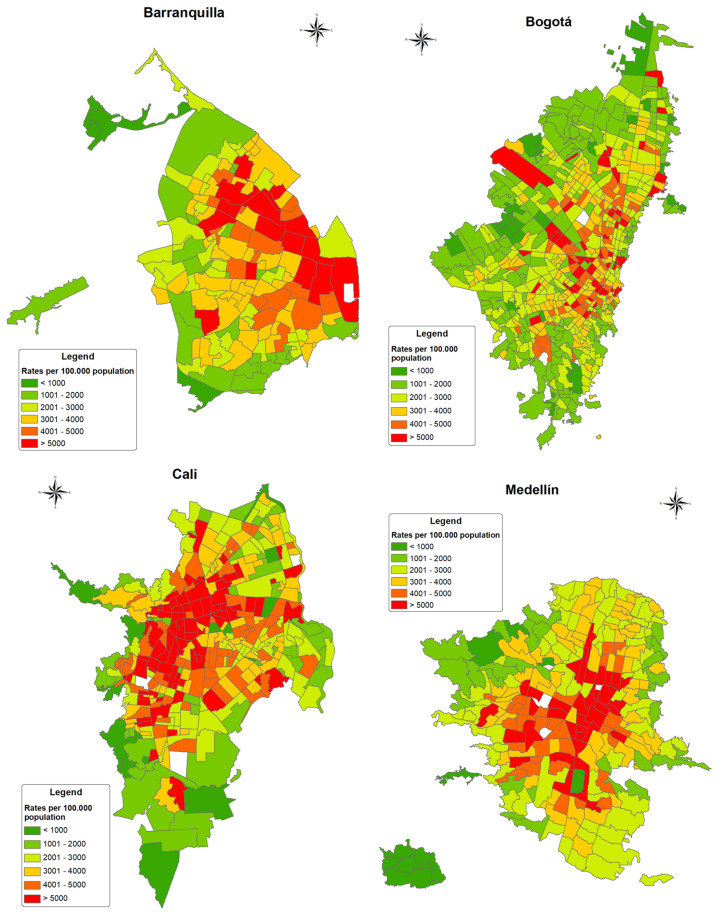
Bayesian mortality rates for all deaths by city in Colombia 2015–2019.

**Figure 2 ijerph-20-00992-f002:**
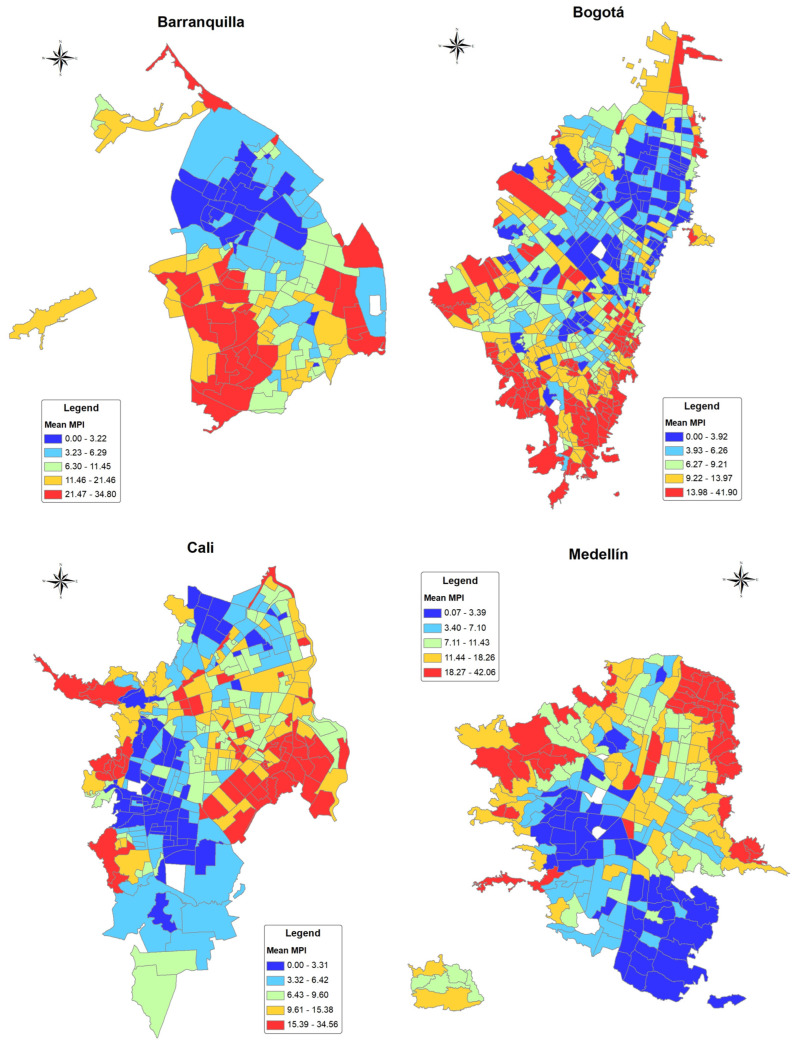
Multidimensional poverty index (MPI) at census sector level by city in Colombia 2018.

**Figure 3 ijerph-20-00992-f003:**
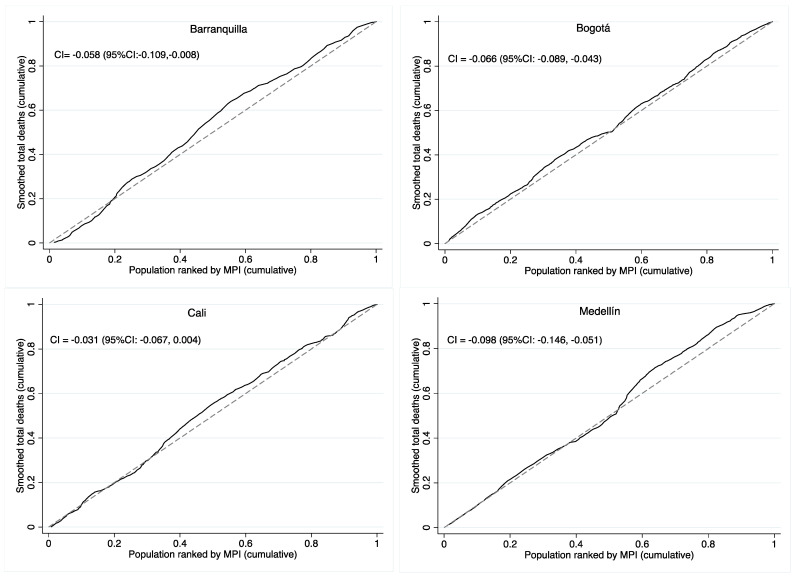
Concentration index (CInd) of mortality for all deaths by multidimensional poverty index (MPI) for four cities in Colombia, 2015–2019.

**Figure 4 ijerph-20-00992-f004:**
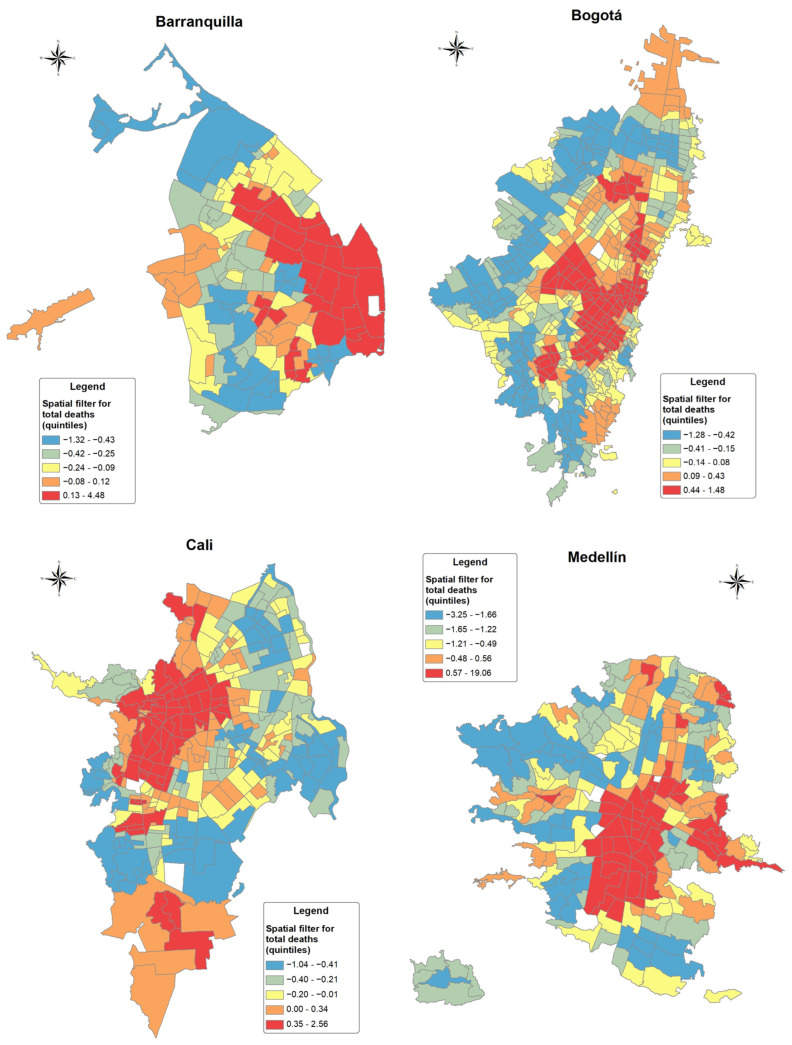
Moran eigenvector spatial filter for all deaths at census sector level by city in Colombia, 2015–2019. Note: Quintiles were split according to the spatial filter range values for each city.

**Figure 5 ijerph-20-00992-f005:**
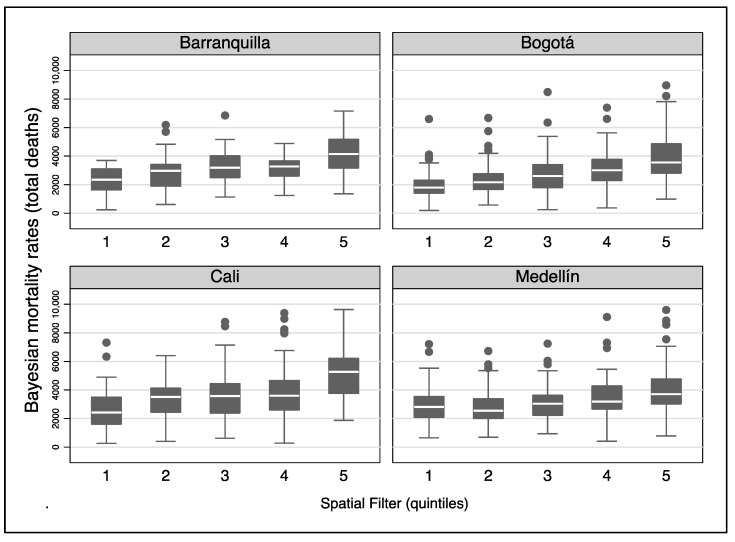
Geographic gradient of Bayesian mortality rates for all deaths across census sectors by the Moran eigenvector spatial filter quintiles by city in Colombia, 2015–2019.

**Table 1 ijerph-20-00992-t001:** Number of deaths and crude mortality rates for all deaths and specific causes of death at the city and census sector level for four cities in Colombia, 2015–2019.

Statistics/City	Barranquilla		Bogotá		Cali		Medellín	
	No.	Rate × 100,000	No.	Rate × 100,000	No.	Rate × 100,000	No.	Rate × 100,000
Total deaths (adults and non-violent)	24,118	3113.28	114,074	2170.06	44,927	3343.41	53,426	3062.21
Total circulatory deaths	9442	1218.82	40,559	771.57	16,245	1208.93	15,608	894.60
Total respiratory deaths	3114	401.97	15,428	293.49	7127	530.38	7295	418.13
Total cancer/blood deaths	5600	722.88	33,128	630.20	12,168	905.53	13,458	771.37
Adult population (19 years and older)	774,681		5,256,715		1,343,749		1,744,689	
No. Census sectors	150		624		340		242	
Statistics by census sector	Min–Max	Median (p25–p75)	Min–Max	Median (p25–p75)	Min–Max	Median (p25–p75)	Min–Max	Median (p25–p75)
No. Deaths	1–812	106.5 (50–247)	1–909	140.5 (70–259)	1–728	97 (45–180)	1–726	209 (106–308)
No. Circulatory deaths	1–299	40 (20–94)	0–323	49.5 (25–93)	0–371	36 (14.5–67)	0–272	57 (26–93)
No. Respiratory deaths	0–114	13.5 (6–28)	0–126	20 (8–36)	0–100	15 (7–31)	0–134	28 (10–46)
No. Cancer/blood deaths	0–181	25 (10–55)	0–314	40 (20–71)	0–189	26.5 (12–49.5)	0–199	53 (28–76)
Mortality rate all deaths × 100,000	28.26–46,649.48	3175.85 (2272.72–3846.15)	96.04–84,210.52	2562.53 (1740.79–3516.81)	72.02–33,333.3	3603.24 (2316.08–4487.16)	370.28–250,000	3088.93 (2349.83–4296.58)
Mortality rate Circulatory × 100,000	28.26–14,175.26	1226.60 (808.63–1632.44)	0–31,578.95	880.75 (580.96–1288.94)	0–16,666.67	1256.03 (762.55–1827.53)	0–35,135.14	919.45 (627.97–1217.04)
Mortality rate Respiratory × 100,000	0–5670.10	401.40 (263.64–569.07)	0–16,666.67	331.23 (213.66–520.45)	0–7055.22	545.36 (303.65–808.99)	0–21,621.62	418.42 (267.93–620.20)
Mortality rate Cancer × 100,000	0–12,121.21	711.38 (489.22–950.12)	0–31,578.95	702.38 (497.12–965.85)	0–16,666.67	981.83 (628.06–1348.62)	0–133,333.3	777.88 (562.71–1105.29)

**Table 2 ijerph-20-00992-t002:** Concentration indexes (CInd) of mortality for all deaths and specific causes of death by multidimensional poverty index (MPI) for four cities in Colombia, 2015–2019.

**Index/City ***	**Barranquilla**		**Bogotá**		**Cali**		**Medellín**	
	**Index**	**95% CI**	**Index**	**95% CI**	**Index**	**95% CI**	**Index**	**95% CI**
**Concentration Index-CInd**								
All deaths	**−0.058**	**−0.109, −0.008**	**−0.066**	**−0.089, −0.043**	−0.031	−0.067, 0.004	**−0.098**	**−0.146, −0.051**
Circulatory deaths	−0.032	−0.079, 0.014	**−0.064**	**−0.089, −0.039**	−0.011	−0.049, 0.027	**−0.107**	**−0.153, −0.062**
Respiratory deaths	**−0.066**	**−0.114, −0.018**	**−0.042**	**−0.067, −0.017**	**−0.052**	**−0.091, −0.013**	**−0.166**	**−0.206, −0.126**
Cancer/blood deaths	**−0.097**	**−0.142, −0.052**	**−0.091**	**−0.110, −0.072**	**−0.064**	**−0.094, −0.033**	**−0.126**	**−0.175, −0.078**
**CI at Extremes-CIE**	**−0.17**	**−0.23, −0.12**	**−0.12**	**−0,15, −0.10**	**−0.18**	**−0.21, −0.14**	**−0.17**	**−0.21, −0.13**

* Statistically significant values in bold.

**Table 3 ijerph-20-00992-t003:** Multivariable Poisson regression models on the association between smoothed Bayesian mortality rates for all deaths and quintiles of multidimensional poverty index (MPI) and concentration index at extremes (CIE) for selected cities in Colombia, 2015–2019.

City-Model Estimations	Models Using MPI						Model Using CIE	
	Model 1		Model 2		Model 3		Model 4	
	IRR	95% CI	IRR	95% CI	IRR	95% CI	IRR	95% CI
**Barranquilla**								
Index-Q1	Reference		Reference		Reference		Reference	
Q2	1.02	0.76–1.38	0.99	0.81–1.21	1.08	0.89–1.30	1.08	0.88–1.30
Q3	0.87	0.62–1.22	1.1	0.89–1.37	1.12	0.94–1.32	1.16	0.97–1.39
Q4	**0.67**	**0.48–0.94**	0.83	0.58–1.18	0.88	0.63–1.23	0.88	0.67–1.15
Q5	0.85	0.64–1.13	1.01	0.64–1.58	0.99	0.67–1.49	1.09	0.82–1.47
>25% adults aged 60 or more years	NA		**1.73**	**1.42–2.10**	**1.65**	**1.40–1.94**	**1.62**	**1.39–1.90**
>48% male sex	NA		1.22	0.81–1.85	1.18	0.81–1.69	1.08	0.87–1.33
Spatial Filter	NA		NA		**1.34**	**1.24–1.45**	**1.34**	**1.24–1.46**
Pseudo R-squared	0.07		0.18		0.41		0.42	
Moran´s index of residuals (*p*-value)	0.225 (0.001)		0.198 (0.001)		0.023 (0.274)		0.023 (0.270)	
**Bogotá**								
Index-Q1	Reference		Reference		Reference		Reference	
Q2	0.98	0.83–1.15	1.19	1.04–1.36	**1.18**	**1.04–1.34**	0.99	0.86–1.31
Q3	0.89	0.76–1.04	1.08	0.93–1.26	**1.15**	**1.00–1.34**	0.96	0.83–1.09
Q4	**0.79**	**0.69–0.92**	0.86	0.73–1.02	0.96	0.80–1.25	**0.85**	**0.73–0.99**
Q5	**0.75**	**0.63–0.88**	**0.8**	**0.67–0.96**	0.9	0.77–1.07	**0.79**	**0.67–0.94**
>25% adults aged 60 or more years	NA		**1.78**	**1.57–2.03**	**1.66**	**1.47–1.87**	**1.51**	**1.34–1.69**
>48% male sex	NA		**1.32**	**1.17–1.48**	**1.23**	**1.10–1.37**	**1.22**	**1.10–1.35**
Spatial Filter	NA		NA		**1.46**	**1.35–1.58**	**1.45**	**1.33–1.57**
Pseudo R-squared	0.04		0.17		0.29		0.27	
Moran´s index of residuals (*p*-value)	0,10 (0.001)		0.085 (0.007)		−0.037 (0.951)		−0.035 (0.942)	
**Cali**								
Index-Q1	Reference		Reference		Reference		Reference	
Q2	0.8	0.63–1.02	0.92	0.76–1.11	0.93	0.79–1.09	0.97	0.81–1.15
Q3	0.9	0.73–1.09	1.09	0.93–1.29	1.09	0.93–1.29	1.09	0.91–1.31
Q4	0.95	0.77–1.16	**1.3**	**1.07–1.57**	**1.23**	**1.03–1.45**	1.22	0.98–1.50
Q5	**0.76**	**0.62–0.96**	**1.19**	**0.98–1.46**	**1.23**	**1.02–1.48**	1.18	0.95–1.46
>25% adults aged 60 or more years	NA		**1.81**	**1.59–2.07**	**1.34**	**1.17–1.53**	**1.31**	**1.13–1.51**
>48% male sex	NA		0.92	0.69–1.23	**0.78**	**0.64–0.95**	**0.81**	**0.67–0.98**
Spatial Filter	NA		NA		**1.71**	**1.55–1.89**	**1.71**	**1.54–1.89**
Pseudo R-squared	0.03		0.2		0.37		0.36	
Moran´s index of residuals (*p*-value)	0.251 (0.001)		0.198 (0.001)		−0.063 (0.980)		−0.070 (0.993)	
**Medellín**								
Index-Q1	Reference		Reference		Reference		Reference	
Q2	1.04	0.87–1.24	1.17	1.02–1.34	**1.19**	**1.03–1.39**	1.02	0.83–1.25
Q3	0.88	0.75–1.02	1.14	1.12–1.67	**1.4**	**1.15–1.69**	1.09	0.89–1.32
Q4	**0.81**	**0.67–0.96**	1.24	0.95–1.61	1.27	0.99–1.64	1.06	0.87–1.30
Q5	**0.62**	**0.52–0.75**	0.85	0.67–1.11	0.95	0.74–1.23	0.79	0.61–1.02
>25% adults aged 60 or more years	NA		**1.91**	**1.53–2.37**	**1.88**	**1.55–2.29**	**1.67**	**1.45–1.90**
>48% male sex	NA		**1.42**	**1.18–1.70**	**1.29**	**1.09–1.53**	**1.31**	**1.08–1.58**
Spatial Filter	NA		NA		**1.06**	**1.02–1.08**	**1.05**	**1.02–1.09**
Pseudo R-squared	0.11		0.31		0.35		0.33	
Moran´s index of residuals (*p*-value)	0.234 (0.001)		0.078 (0.001)		−0.022 (0.720)		−0.014 (0.606)	

## Data Availability

Not applicable.

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
