# Peer review of "Intraurban Geographic and Socioeconomic Inequalities of Mortality in Four Cities in Colombia"

_ijerph, 2023, doi:10.3390/ijerph20020992_

Round 1

Reviewer 1 Report

Manuscript: ijerph-2080668

Title: Intraurban geographic and socioeconomic inequalities of mortality in four cities in Colombia.

Thank you very much for the opportunity to read and review the manuscript. This study is interesting, important and needed in the literature to make visible the environmental determinants in countries of Latin America. However, several clarifications need to be made before considering for publication.

Comment 1. Abstract: What is a “Bayesian mortality rate”?

Comment 2. The acronym CI is normally used for Confidence Intervals, I suggest that the authors use a different acronym for Concentration Indexes to avoid confusion 

Comment 3. The abstract defines MPI as Mortality Poverty Index, is this a typo? In the methods, you defined MPI as Multidimensional Poverty Index. Please clarify. If the abstract is correct, then it sounds like you are explaining spatial patterns of mortality with quintiles of a mortality index, this is confusing, please clarify. 

Comment 4. Consider adding a map including the 4 cities highlighted so that it is easier for the reader to localize them.

Comment 5. How did authors adjust for variability in population size across CS and across cities? 

Comment 6. The authors mention that one dimension of the MPI is health. Can you specify what health metrics are considered? If mortality is one of them, this would induce milti-correlation and thus bias in the results. 

Comment 7. Please clarify what causes of death with codes 140 ICD-10 S00-T98 and V01-Y98 are excluded.

Comment 8. How confident are the authors that the BMRs represent the actual MRs? Since this is the main outcome, I was expecting some validity analysis or further information on generalizability, internal validity, etc. 

Comment 9. Did the authors evaluate nonlinear association between BMR and MPI? Why go stratight to quintiles? 

Comment 10. In Model 4, did the main exposure get switched to ICE? This was not clear. 

Comment 11. The sample size is missing in the Methods and Study population section, Data Sources, Socioeconomic and population data section, and line [128],[129].

Comment 12.  line [90],[91] Please clarify why data for population younger than 18 years were excluded.

Comment 13. In “Definition of geographical areas” section, the authors mentioned that they used cartographic information for the year 2018. Nevertheless, my concern is regarding the rest of the follow-up period. Please clarify why did the authors did not include data from 2015, 2016, and 2019 respectively. 

Comment 14. line [124] Same comment as above

Comment 15. line [154] The construction of the “Descriptive mortality analysis” is more interesting. However, I would have liked to see the formula more as a graphical equation than in text. This is more friendly visually.

Comment 16. Can the authors expand on outcome misclassification? Especially if the data source only uses the registered place of death and may miss the mobility and/or change of residence of the participant during the lifetime. 

Comment 17. I understand that the nature of the study, as well as the exposure and outcome variables, make it difficult to obtain more information about the selection of the confounders (eg. Medical treatment, diet, lifestyle, physical activity, etc.). However, I would like to know if the authors contemplated bias due to missing confounders and how did they adjust for this. 

Comment 18. This comment is regarding the format of the sections of the manuscript (Mainly in Methods). For the consideration of the authors, I suggest numerical sequence of subscripts and sections throughout the document (eg. 2.1, 2.2, 2.3, 2.4….). 

lines [387-391] I suggest moving these lines to the methods, to the corresponding section. Finally, I think there is a format problem and table 1 is not complete, I suggest adjusting it to the document, the data for Medellin cannot be seen.

Author Response

Please find responses to reviewers in attached file.

Reviewer 2 Report

The work establishes a clear objective, has a congruence between the design and the methodology used.

The calculation of the Moran index is an interesting and adequate proposal for the purposes of the study.

The decision to adjust the mortality considering the population size of the units of analysis allows for a more adequate comparison.

The results in Table 1 cannot be reviewed since it is incompletely presented in its pdf file presentation.

Table 2, statistically significant values are identified in bold. The presentation of this table should be reviewed since there are significant results that are not in bold, as in the case of respiratory deaths in Barranquilla. 

Author Response

(The authors gave the same response as above.)
